# Perianal Structures in Non-Myrmecophilous Aphids (Hemiptera, Aphididae)

**DOI:** 10.3390/insects14050471

**Published:** 2023-05-16

**Authors:** Natalia Kaszyca-Taszakowska, Mariusz Kanturski, Łukasz Depa

**Affiliations:** Institute of Biology, Biotechnology and Environmental Protection, Faculty of Natural Sciences, University of Silesia in Katowice, Bankowa 9, 40-007 Katowice, Poland; mariusz.kanturski@us.edu.pl (M.K.); lukasz.depa@us.edu.pl (Ł.D.)

**Keywords:** insects, ecology, mutualism, morphology, adaptation, ants

## Abstract

**Simple Summary:**

We studied perianal structures in 25 species of non-myrmecophilus aphids representing five subfamilies. In our research, we focused on the thoracic segment VIII, cauda, anal plate, and genital plate their microstructure and setae. We compared the results of measurements of these structures with the results of facultatively and obligatory myrmecophilous aphids. Our observations show that non-myrmecophilous and myrmecophilous aphids form two clusters, relatively well separated and following the two important morphological features: the length of the anal plate and the length of the cauda in relation to the length of the anal plate. It seems that myrmecophilous aphids may have long cauda, but it must be shorter than the height of the anal plate. However, the anal plate must be higher than its width. These studies confirmed the theory of differences in the structure of perianal structures in aphids depending on the relationship with ants and confirm the existence of the trophobiotic organ.

**Abstract:**

Mutualistic relation with ants is one of the leading features of aphid ecology. For some aphid species, it is a crucial association enhancing their survival capability, while the life mode of some others is completely independent of ants. It was broadly accepted that during the evolution of aphids, the ones relying on ants developed special morphological adaptations for this mutualism, the so-called trophobiotic organ. Its exact structuring, however, posed some explanatory difficulties because many non-myrmecophilous aphids had structural modifications accordant with the trophobiotic organ, while some myrmecophilous did not. Here we present an evaluation of the morphology of perianal structures in 25 non-myrmecophilous aphid species with reference to previous, similar studies on myrmecophilous species based on scanning electron microscopy. We conclude that the trophobiotic organ is an existing adaptation, but its definition requires revision.

## 1. Introduction

Behavior on the ant-aphid line is customarily divided into mutualistic relationships (specified into obligatory and facultative ones) [1] and predation [2,3]. Ant-aphid interaction is a well-known model of mutualism: ants protect the aphids against natural enemies such as predators, ladybugs or lacewings, and parasitoids (e.g., Aphidiinae (Hymenoptera: Ichneumonoidea: Braconidae)) [4], and in return, they receive high-energy honeydew from the aphids [5,6,7]. However, the relationship itself is not so simple; mutualism is affected by many factors, such as aphid population size [6,8], competition between aphids for ants [9], ant species [10], host plant quality [11], ant distribution [12], and the nutritional needs of the ants [13]. However, as noted by Stadler and Dixon [14], increased honeydew production implies an enhanced feeding rate, which could have an adverse effect on assimilation [15] and account for the poorer individual performance of this aphid, i.e., the aphids are buying protection. Once an obligate relationship develops, the aphid does less well at the individual and population levels when not ant-attended. Therefore, it is impossible to unequivocally say which species of aphids are more beneficial, those attended or not attended by ants, and also which pair of mutualists is in obligate and which in facultative relation. 

There is a hypothesis showing the correlation of honeydew composition with the mutual behavior of aphids toward the ant protector [2], and that ants prefer species that give large amounts of honeydew rich in amino acids and/or di- and trisaccharides [16]. It should also be noted that this relationship tends to shift for hosting ants. As the aphid colony grows, their protection is too costly for the ant hosts, which reduces the number of aphids while providing a source of protein [17,18,19]. If we consider the search for morphological features of myrmecophily, it was noted that species of aphids feeding on plants with woody stems have a longer stylet, which they insert into the phloem of the host plant tissue. Therefore, due to reduced mobility and the chance for quick defense mechanisms (such as involuntarily falling from the host), mutualism with ants would be preferable for them [20]. However, this may be partly questioned because while, e.g., *Stomaphis* Walker aphids feeding on tree trunks indeed have a long rostrum and depend on ants [21], the *Phloeomyzus passerinii* aphids also feed on the tree trunk and have relatively long rostrum. Still, ants do not attend it at all [22]. The costs and benefits of aphid–ant interaction are difficult to measure. In early studies of the interaction of ants with aphids, only benefits were reported, but we now assume that there is always a scale of costs and benefits. There are many parameters taken into account, e.g., endosymbionts operating within the aphids [23,24,25], intra-guild predation between different predators [26], and selective removal of parasitic aphids by grooming ants [27,28]. 

In view of the above-mentioned mutual influences in ant-aphid interactions, one of the open issues remains morphological adaptations to mutualism in aphids, especially the existence of the so-called trophobiotic organ. The concept of this organ was set by Mordvilko [22], and it was later repeatedly applied in studies on aphid biology and ant-aphid mutualism. It assumed that myrmecophilous aphids have vertical anal plates and short cauda, while the anal pore is surrounded by a ring of elongated setae serving to keep the honeydew droplet until it is taken by ant worker. The first review of this concept was done by Kanturski et al. [29], but it followed Zwölfer’s approach (following Mordvilko), focusing only on subterranean, myrmecophilous aphids. It confirmed the existence of certain similarities in myrmecophilous aphids linked to the trophobiotic organ, e.g., vertical anal plate, long setae around the anus, and short cauda. However, myrmecophily was not the only common feature of the studied species. Short cauda was a common feature of higher taxa to which the studied species belonged (Eriosomatinae *s. lat*., Anoeciinae, Lachninae), as well as constant or temporary subterranean life mode, additional to, e.g., living in galls (Fordini, e.g., genera *Forda*, *Geoica*). This gave starting points to the evaluation of the concept of the trophobiotic organ because it was not clear whether observed morphological structures are adaptations to myrmecophily or, perhaps, to subterranean life mode or living in confined space in galls. The review of the morphology of the perianal region in representatives of the obligatory myrmecophilous genus *Stomaphis* confirmed its similarity to *Trama* (also Lachninae) [21]. Additionally, it reexamined peculiar perianal tubercles near the anus, having harshly protuberant cuticular denticles, which were hypothesized to serve as protection against sticky honeydew. The structure of the cuticle on these tubercles resembled the structuring of aphid cauda, which is regarded as an organ protecting from honeydew, thus suggesting that the structuring of the cuticle surface may have an important role in supporting the protection against honeydew droplets, e.g., until it is removed by ants (a suggested role of the trophobiotic organ).

However, the review through facultative and obligatory myrmecophilous aphids [1] still did not resolve the problem of the existence of trophobiotic organs, as it was clear that both ways of engagement of aphids into mutualism with ants are correlated with long cauda and long perianal setae. Within the subfamily Aphidinae, the obligatory myrmecophilous aphids have even longer and more slender cauda than facultative ones. The current paper follows the conducted research so far, focusing on non-myrmecophilous aphids. We wanted to check if there are any differences in the morphology of perianal structures and microstructures that prevent or hinder the transfer of honeydew to an ant. We also tried to check whether there are differences in the structure of the perianal region between obligately and facultatively myrmecophilous [1] and non-myrmecophilous aphids, including generations living in galls.

## 2. Material and Methods

### 2.1. Taxon Sampling

All aphid species were collected from Poland and Israel from September 2019 to September 2022. Each species was collected into a set of tubes containing 70% ethanol (EtOH) for slide preparation and scanning electron microscope. *Baizongia pistaciae* was collected by Moshe Inbar in Israel from *Pistacia* sp.

The list of species studied in this research is presented in Table 1. All of the 25 aphid species studied here are non-myrmecophilous. In the case of representatives of the subfamily Eriosomatinae, the study focused on alate females feeding in galls on the primary host (not attended by ants). A total number of 25 species were studied, including 21 with scanning electron microscopy and light microscopy and four only with light microscopy, which resulted from unsatisfactory preservation of specimens of these species mounted for SEM.

### 2.2. Material

After mounting on the microscopic slide with the method described by Kanturski & Wieczorek [30], the specimens were identified using keys by Blackman & Eastop [31], Wojciechowski et al. [32], and Heie [33,34,35,36,37,38,39]. The taxonomic system was applied according to Favret [40]. Adult female aphids, both apterae and alatae, were used for this study, and efforts were made to collect a minimum of three mature specimens.

### 2.3. Scanning Electron Microscopy

Scanning electron microscopy (SEM) was used as the primary method in this study (Figures 1–6). Depending on the availability of the material, two to seven individuals have been fixed and analyzed using SEM. For preparation, the SEM studies method used by Kanturski et al. [29] was applied, but it was modified (samples were collected in test tubes with 70% alcohol and the phosphotungstic acid (PTA) stage was omitted) as follows.

The EtOH samples were prepared following standard protocol: 80% ethanol (five min), 90% ethanol (five min), 96% ethanol (about ten min. each), and two changes of absolute ethanol for 15 min each. After this, samples were put into plastic tubs, and chloroform was added for 48 h. In the next step, we used hexamethyldisilazane (HMDS) solution with EtOH in the proportion of 1/3 for 20 min, 1/2 for 30 min, 2/3 for 45 min, and 3/3 for 30 min (2 × 15 min). Dry samples were glued to 32 mm diameter tables with carbon discs in different positions, allowing a detailed analysis of the species from different perspectives. Exceptions were species with a small number of preserved specimens, in which case we tried to make the picture as full as possible of the aphid’s entire anal structure. Aphids were sputtered with a 30 nm layer of gold. SEM micrographs were obtained using a Phenom XL scanning electron microscope (Phenom-World B.V., Eindhoven, Netherlands) at 15 kV accelerating voltage with a BackScatter Detector (BSD). The figures were prepared using the Adobe Photoshop CS6 graphic editor. The perianal structures were described in the figures as follows: tergite VIII (VIII t); cauda (cd); anal pore (ap); anal plate (apl); genital plate (gpl); genital pore (gp).

### 2.4. Morphological Measurements

Perianal structure measurements (Appendix A) were measured from microscopic slides with a Leica DM 3000 upright light microscope with a Leica MC 190 HD digital camera and LasX ver. 5.1.025593 software and partly from SEM. The detailed measurements and ratios were established following the methods presented by Kaszyca-Taszakowska et al. [1]. Also, for comparison of non-myrmecophilous and myrmecophilous aphids, the source data on perianal structures of 17 obligatory myrmecophilous and 12 facultatively myrmecophilous aphid species presented by Kaszyca-Taszakowska et al. [1] were applied (Figure 7). For statistical analysis of the obtained values, the Shapiro–Wilk test for distribution type was performed (with W = 0.9818 > W (α = 0.05, n = 54) = 0.957), and for the difference between the ratios, standard *t*-test was performed for unpaired samples, with *p* < 0.05. For statistical analysis, Statsoft Statistica 13.3 (StatSoft, Inc., Tulsa, OK, USA (2011)) software was applied.

## 3. Results

A total number of 25 species was analyzed, belonging to five subfamilies (Table 1). The details of the morphology of the perianal structures of these species are presented in Figures 1–6. Abbreviations for morphological structures are listed in the M&M section. The order of species does not strictly follow systematic classification, but the degree of resemblance of the perianal region to so far defined trophobiotic organ, e.g., starting with species with very long cauda through perianal regions variously structured and ending with species with short cauda.

### 3.1. Morphological Characteristics of Perianal Structures

*Macrosiphoniella artemisiae* (Boyer de Fonscolombe, 1841)

(Figure 1a–c).

**Abdominal tergite VIII:** possesses a fine microsculpture and a few short setae with pointed apices at the cauda base are present.

**Cauda:** large, elongated, reaching up to 450 micrometers, with a surface relief slightly more significant on the ventral side. Setae are on both sides of the cauda and the apex. 

**Anal plate:** slightly emarginate, square, with four setae in the middle, and a few setae on both sides of the plate. In the center of this area, the microstructure is a little bit more massive and larger. 

**Genital plate:** a large, almost smooth structure, with a row of short, massive setae right at the posterior margin, near the genital pore, and four–six longer setae at the anterior edge.

*Macrosiphoniella pulvera* (Walker, 1848).

(Figure 1d–f).

**Abdominal tergite VIII:** possesses an almost smooth surface, setae set towards the anal plate, occurring on the sides of the segment (they are absent in the middle where the cauda is located below).

**Cauda:** elongated, with a more pronounced microstructure than on segment VIII, with slightly longer cuticle processes on the ventral side (towards the anal pore). Setae on both sides of the cauda and on the outer side in its apical part.

**Anal plate:** slightly emarginate, with setae turned upwards, with two distinct setae in its central part and slightly shorter ones on both sides.

**Genital plate:** large, rectangular; thickened stripe of cuticular microstructure on the posterior edge at the genital pore, with a row of setae just below. 

*Uroleucon aeneum* (Hille Ris Lambers, 1939).

(Figure 1g–i).

**Abdominal tergite VIII:** with a delicate texture of cuticle and four setae ending with a club.

**Cauda:** elongated, more than 300 micrometers long. Long setae are located unregular on both sides of the cauda, appearing from about the middle of its length—where the range of the setae with the anal plate ends. The apex of the cauda has a higher density of setae over the entire surface. Setae are shorter the closer to the apex, with a small cube. The cuticular microstructure on the cauda is clear and uniform over the entire surface.

**Anal plate:** slightly emarginate, in the central part, it forms a semicircle with thick setae, shorter than the lateral setae on the cauda. In the center of this area, the microstructure is more massive and larger; in the rest of the anal plate, it is delicate and increases its intensity towards the genital plate.

**Genital plate:** possesses a very delicate microstructure; setae located on the edge near the genital pore, and two large setae in the central part of the plate, facing upwards. Sometimes, there are a few smaller setae near them.

**Figure 1 insects-14-00471-f001:**
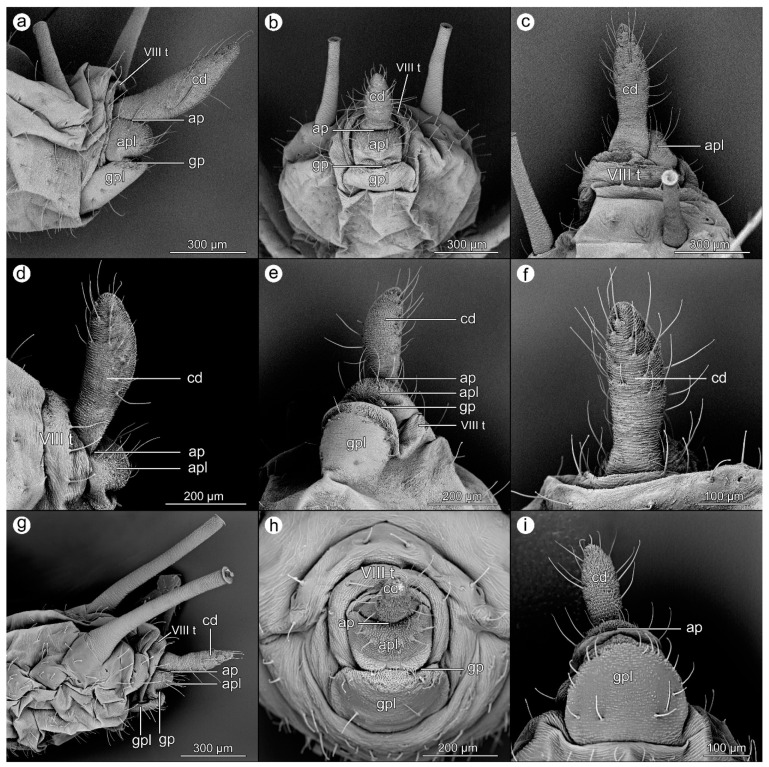
Perianal structures of *Macrosiphoniella artemisiae*: (**a**)—lateral view, (**b**)—rear view, (**c**)—dorsal view; *Macrosiphoniella pulvera*: (**d**)—lateral view, (**e**)—ventral view, (**f**)—dorsal view; and *Uroleucon aeneum*: (**g**)—lateral view, (**h**)—rear view, (**i**)—ventral view.

*Uroleucon jaceae* (Linnaeus, 1758).

(Figure 2a–c).

**Abdominal tergite VIII:** microstructure of cuticle in the form of rows of minute denticles.

**Cauda:** elongated, with erect setae on both sides; microsculpture with increasing harshness towards the apex. 

**Anal plate:** rectangular, the cuticular microrelief is distinct, with longer cuticle outgrowths located in the midline and towards the genital pore. Setae are grouped into two groups on either side of the anal plate.

**Genital plate:** large, strongly sclerotized, with a fine microsculpture and the most setae in line next to the genital pore and few in its anterior part. 

*Cavariella pastinacae* (Linnaeus, 1758).

(Figure 2d–f).

**Abdominal tergite VIII:** microsculpture typical of the rest of the body, consisting of flat tubercles with only minimal denticles towards the anus and with a single spinal process ended with two setae and covered by flat, rather smooth imbrications. 

**Cauda:** broadly finger-like, ventrally covered by harsh cuticular denticles and a few relatively short setae laterally placed and bent towards the long axis of the body and anal pore.

**Anal plate:** wide and short, covered with very protuberant cuticular denticles and long, erect setae. 

**Genital plate:** wide and long, rectangular with rounded corners; smooth with delicate microscuplture only in the anterior part. Few short setae on the anterior and posterior margins. 

*Corylobium avellanae* (Schrank, 1801).

(Figure 2g–i).

**Abdominal tergite VIII:** covered with delicate cuticular microsculpture and long, erect setae ended with a small club. 

**Cauda:** broadly finger-like, slightly raised upwards, ventrally covered by sharp and prominent cuticular denticles and a few, relatively short setae laterally placed and bent towards the long axis of the body and anal pore. 

**Anal plate:** wide and short, covered with very protuberant cuticular denticles along the anal pore and long, erect setae.

**Genital plate:** wide and long, rectangular with rounded corners; smooth with delicate microscuplture near the margins. Posterior and lateral margins with short, erect setae, while setae on the anterior margin are longer, all of them blunt. 

**Figure 2 insects-14-00471-f002:**
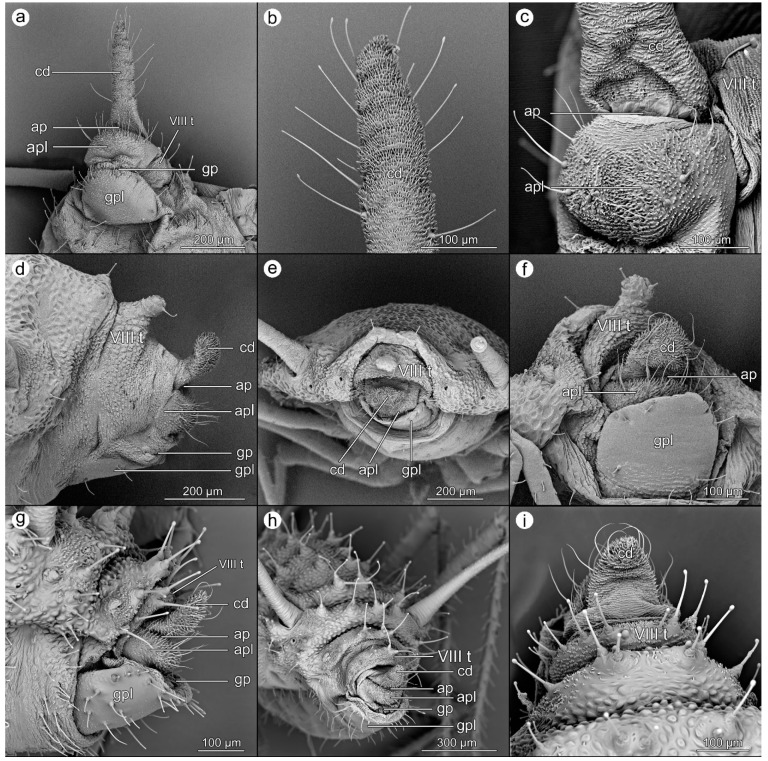
Perianal structures of *Uroleucon jaceae*: (**a**)—rear view, (**b**)—close-up on the surface of cauda, (**c**)—close-up on the anal pore and anal plate; *Cavariella pastinacae*: (**d**)—lateral view, (**e**)—rear view, (**f**)—ventral view; and *Corylobium avellanae*: (**g**)—lateral view, (**h**)—oblique view, (**i**)—dorsal view.

*Drepanosiphum platanoidis* (Schrank, 1801).

(Figure 3a–c).

**Abdominal tergite VIII:** without setae, covered by transverse rows of minute cuticular spinules.

**Cauda:** short, finger-like; covered with sparsely distributed single cuticular denticles, more prominent on the ventral side and with few long setae also on the ventral side, bent towards the anal pore. 

**Anal plate:** wide and short, covered with sparsely distributed cuticular denticles, either single or in rows; setae long and bent towards the anal pore, arranged in two groups on not prominent domes. 

**Genital plate:** oval, rather smooth, with single cuticular denticles on the anterior part, and two rows of setae bent towards the genital pore on its posterior part. 

*Euceraphis betulae* (Koch, 1855).

(Figure 3d–f).

**Abdominal tergite VIII:** covered with transverse rows of spinules on its distal margin. 

**Cauda:** with knob-like apex and conical base; dorsal part of base and apex covered with rows of the serrated cuticle, while ventral part covered with single cuticular denticles. A few short setae are on the dorsal surface of the knob, while several long and bent setae are on the apical and ventral parts of the knob. 

**Anal plate:** wide and short, domed, with cuticle generally smooth but with single cuticular denticles sparsely distributed on lateral margins and more prominent denticles more densely covering the central part of the plate. Setae are long and slightly bent towards the anal pore, arranged in two groups along lateral parts of the plate. 

**Genital plate:** wide, smooth, with delicate cuticular microsculpture mainly present along the distal margin, near the genital pore; Setae cover the distal margin of the plate, longer on the lateral margin while shorter near the genital pore and bent towards it. 

*Clethrobius comes* (Walker, 1848).

(Figure 3g–i).

**Abdominal tergite VIII:** cuticular microsculpture is very delicate, with an almost smooth surface and barely distinct denticles and serrations; setae are long, reaching half cauda length.

**Cauda:** knob-like apex with a conical base, but the apex only weakly separated from the base; dorsal part of base and apex covered with rows of the serrated cuticle, while the ventral part is covered with sets of a few cuticular denticles. Many long and bent setae are on the apical and ventral parts of the knob.

**Anal plate:** wide and short, domed, with single cuticular denticles sparsely distributed on lateral margins and more prominent denticles more densely covering the central part of the plate. Setae are numerous, long, and slightly bent towards the anal pore, arranged in two groups along lateral parts of the plate. Under the anal plate, there are clusters of long setae of rudimentary gonapophyses bent towards the genital pore. 

**Genital plate:** wide, smooth, with delicate cuticular microsculpture present all over its surface; long setae also cover the whole plate. 

**Figure 3 insects-14-00471-f003:**
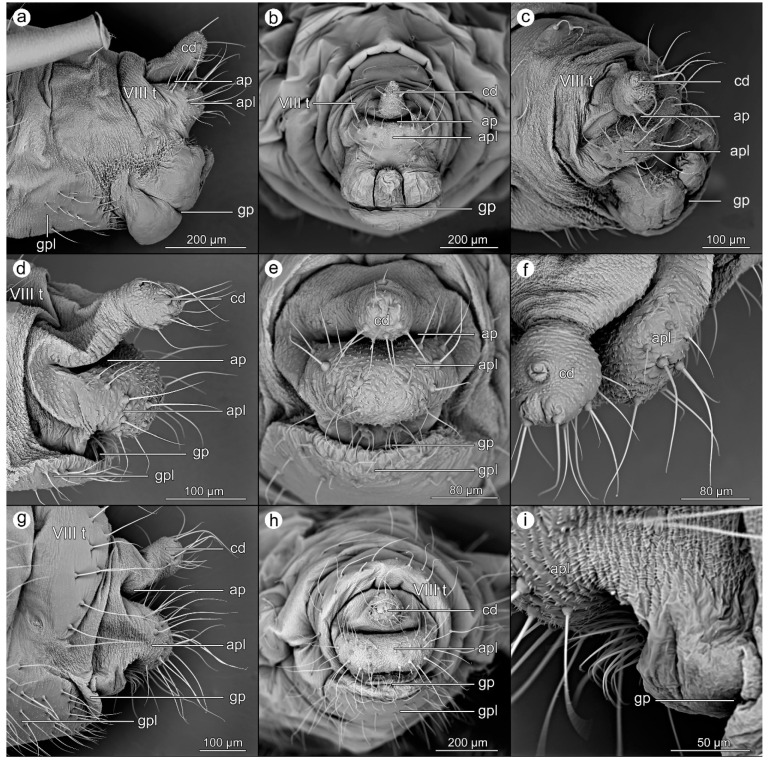
Perianal structures of *Drepanosiphum platanoidis*: (**a**)—lateral view, (**b**)—rear view, (**c**)—oblique view; *Euceraphis betulae*: (**d**)—lateral view, (**e**)—rear view, (**f**)—close-up on the surface of cauda and anal plate; and *Clethrobius comes*: (**g**)—lateral view, (**h**)—rear view, (**i**)—close-up on the surface of anal plate and genital pore.

*Myzocallis carpini* (Koch, 1855).

(Figure 4a–c).

**Abdominal tergite VIII:** cuticular microsculpture is very delicate, with an almost smooth surface and barely distinct denticles; fine setae bent towards the cauda. 

**Cauda:** knobbed, with apex well separated from the base; cuticular microscuplture of the apex in the form of sets of sharp denticles, arranged into transverse rows while the base of cauda with serrated rows on the dorsal side. The entire apex has long, erect setae, some of them bent towards the long axis of the body.

**Anal plate:** wide and short, with two prominent lobes laterally positioned. While the base of the anal plate is relatively smooth, the cuticle of the lobes is covered with rows of sharp denticles, similar to cauda. The lobes are also covered with long, erect setae, with some of them bent towards the long axis of the body. At the lower margin of the genital pore, the two clusters of setae on small tubercles are well visible—the rudimentary gonapophyses. 

**Genital plate:** longer than wide, covered with delicate, serrated rows of cuticles. The posterior margin, near the genital pore, with a row of short setae bent towards the pore, while lateral parts of the plate with few longer setae.

*Prociphilus fraxinifolii* (Riley, 1879).

(Figure 4d–f).

**Abdominal tergite VIII:** cuticle with serrated rows and multi-faceted wax gland plates with single seta rising from their center.

**Cauda:** very short, with a few setae and delicate serration of cuticle.

**Anal plate:** prominent, long, and obliquely positioned; cuticle smooth with rows of lateral serration, while middle part only with single, small denticles. Many long setae, with some bent upwards, toward the anal pore. 

**Genital plate:** rectangular, rather smooth, with rows of cuticular serration at its anterior part and only small, single denticles on its posterior part, near the genital pore. Posterior margin with row of setae bent towards the pore, and the remaining part of the plate with single, erect setae.

*Tinocallis takachihoensis* (Higuchi, 1972).

(Figure 4g–i).

**Abdominal tergite VIII:** cuticular microsculpture is very delicate, with an almost smooth surface and barely distinct denticles; fine setae bent towards the cauda. 

**Cauda:** knobbed, with apex well separated from the base; cuticular microscuplture of the apex in the form of single or sets of short denticles, while the base of cauda with serrated rows on the dorsal and lateral sides while its ventral side is similar to the apex. The entire apex has long, erect setae, some of them bent towards the long axis of the body. 

**Anal plate:** wide and short, with two prominent lobes laterally positioned. While the base of the anal plate is relatively smooth, the cuticle of the lobes is covered with single or clusters of short, acute denticles, similar to cauda. The lobes are also covered with long, erect setae, with some of them bent towards the long axis of the body. At the lower margin of the genital pore, the cluster of setae on small tubercles is well visible—the rudimentary gonapophyses. 

**Genital plate:** longer than wide, covered with delicate, serrated rows of the cuticle. The posterior margin, near the genital pore, with a row of long setae, bent towards the pore while remaining part of the plate with few longer setae. 

**Figure 4 insects-14-00471-f004:**
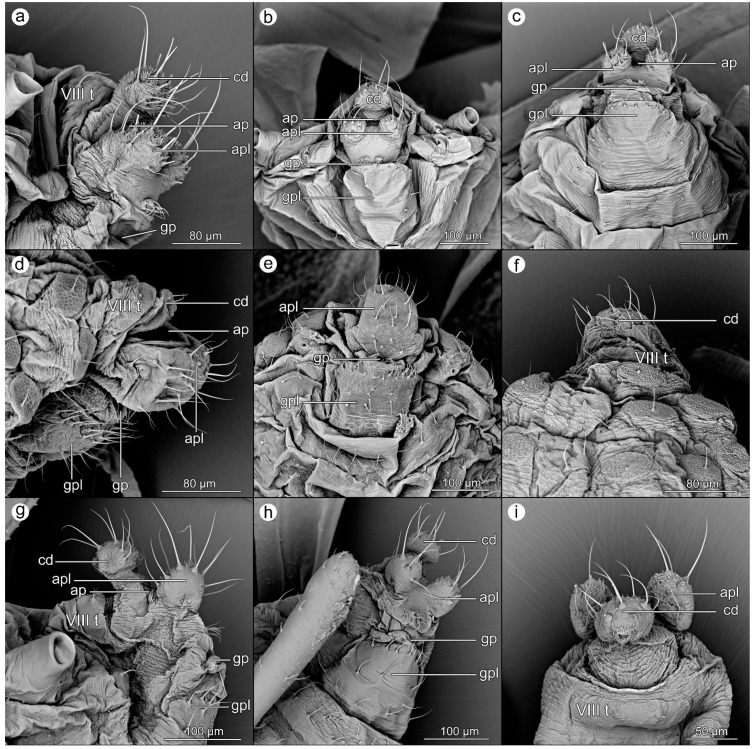
Perianal structures of *Myzocallis carpini*: (**a**)—lateral view, (**b**)—rear view, (**c**)—ventral view; *Prociphilus fraxinifolii*: (**d**)—lateral view, (**e**)—ventral view, (**f**)—dorsal view; and *Tinocallis takachihoensis*; (**g**)—lateral view, (**h**)—ventral view, (**i**)—dorsal view.

*Baizongia pistaciae* (Linnaeus, 1767).

(Figure 5a–c).

**Abdominal tergite VIII:** cuticle relatively smooth, with transverse rows of minute serration and small wax glands laterally placed. A few short setae bent backward. 

**Cauda:** short, with a smooth cuticle and very delicate serrated rows, and a few short setae bent towards the anal pore.

**Anal plate:** longer than wide, with very smooth cuticle and short setae in two lateral rows and along the lower margin, bordering with two gonapophyses in the form of clusters of longer setae on two plates. 

**Genital plate:** small, smooth with weakly serrated rows of cuticles and many short, erect setae close to the genital pore. 

*Eriosoma ulmi* (Linnaeus, 1758).

(Figure 5d–f).

**Abdominal tergite VIII:** cuticle is relatively smooth, with transverse rows of minute lateral serration and few small dorsal denticles. A few short setae bent backward. 

**Cauda:** short, semicircular; cuticular microscuplture in the form of single or sets of short denticles, with few short setae bent toward the anal pore.

**Anal plate:** wide, with smooth cuticle but with small denticles or serrated imbrications, with denticles directed toward the anal pore. Several short setae are mostly laterally placed, slightly bent towards the long axis of the body. 

**Genital plate:** wide, covered with transverse rows of serration and a few irregular rows of relatively short setae directed toward the genital pore. 

*Mimeuria ulmiphila* (Del Guercio, 1917).

(Figure 5g–i).

**Abdominal tergite VIII:** cuticle with transverse rows of serrated imbrications, wax glands, and few long setae. 

**Cauda:** short, semicircular, with smooth cuticle covered by sparsely distributed short denticles, laterally arranged in short rows. Two long setae bent towards the anal pore. 

**Anal plate:** semicircular, protruding, and domed, with transverse rows of serrated cuticular imbrications, better developed along the midline of the plate. All surfaces are covered with many long setae bent towards the long axis of the body. 

**Genital plate:** rectangular, smooth but with very short rows of serrated imbrications, with many long setae sparsely distributed all over its surface, bent towards the genital pore. 

*Tetraneura ulmi* (Linnaeus, 1758).

(Figure 5j–l).

**Abdominal tergite VIII:** cuticle is covered with transverse rows of short denticles and few long setae directed backward. 

**Cauda:** semicircular, covered with transverse rows of short denticles and a few long setae bent toward the anal pore. 

**Anal plate:** rectangular, slightly wider than long, vertically placed; cuticular microsculpture consisting of short, single, acute denticles or spines, sparsely distributed in the mid and upper part of the plate, more abundant in the lateral and lower part. The setae in the upper part are long, bent toward the anal pore, and in the lower part, more abundant, bent toward the genital pore. 

**Genital plate:** semicircular, microsculpture consisting of short, single, acute denticles or spines, sparsely distributed in the anterior part of the plate and more abundant and prominent in the posterior part, close to the genital pore. The posterior part of the plate is covered with many long setae bent toward the genital pore. 

**Figure 5 insects-14-00471-f005:**
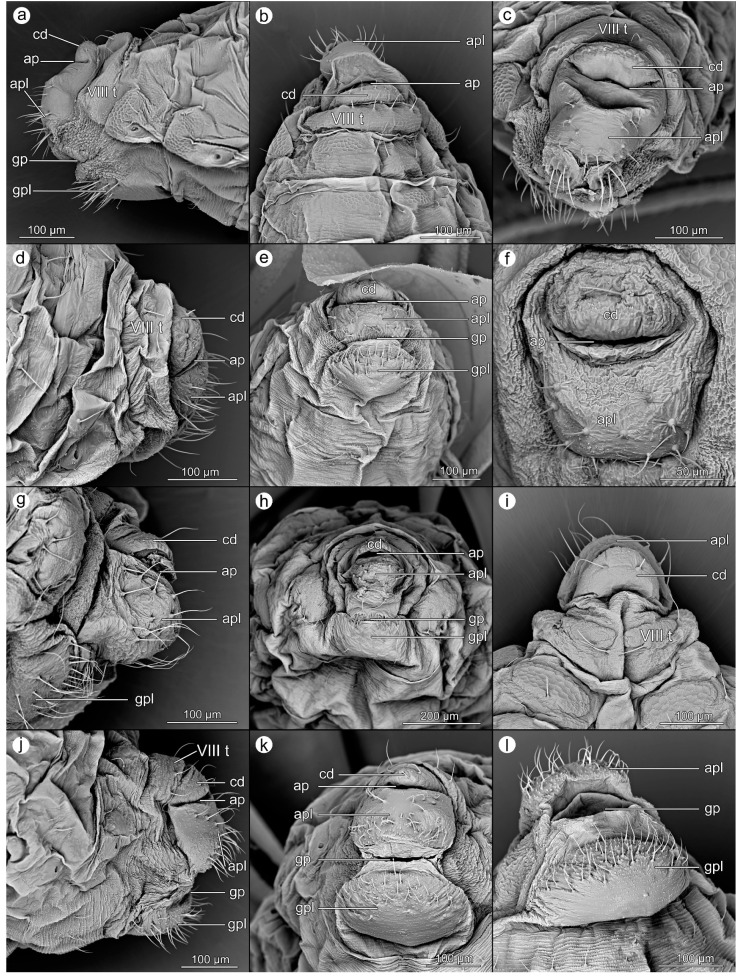
Perianal structures of *Baizongia pistaciae*: (**a**)—lateral view, (**b**)—ventral view, (**c**)—rear view; *Eriosoma ulmi*: (**d**)—lateral view, (**e**)—ventral view, (**f**)—rear view; *Mimeuria ulmiphila*: (**g**)—lateral view, (**h**)—rear view, (**i**)—dorsal view; and *Tetraneura ulmi*: (**j**)—lateral view, (**k**)—rear view, (**l**)—ventral view.

*Liosomaphis berberidis* (Kaltenbach, 1843).

(Figure 6a–c).

**Abdominal tergite VIII:** smooth cuticle, with barely marked serrated imbrications, mostly laterally and posteriorly, and few short, blunt setae. 

**Cauda:** finger-like, covered with transverse rows of prominent, cuticular denticles and a few long setae laterally placed, bent toward the apex of the cauda. 

**Anal plate:** wide, short, covered with many prominent cuticular, acute denticles, and many long, erect setae slightly bent toward the anal pore. 

**Genital plate:** oval, elongated, longer than wide; cuticle is smooth with single rows of small denticles on anterior and posterior margins. A few short setae on the posterior margin, near the genital pore, and two longer, erect setae on the anterior part. 

*Brevicoryne brassicae* (Linnaeus, 1758).

(Figure 6d–f).

**Abdominal tergite VIII:** cuticle covered with transverse rows of serrated imbrications and a few short setae bent backward.

**Cauda:** triangular, covered with sharp denticles, single or arranged in sets, more prominent on the ventral side of the cauda, near the anal pore. A few long setae are laterally placed and bent toward the body’s long axis. 

**Anal plate:** wide, covered with prominent, sharp cuticular denticles on the central area and near the anal pore, and serrated imbrications on lateral margins. Several long setae are bent toward the anal pore.

**Genital plate:** wide, oval, and rather smooth, but with serrated cuticular imbrications in the anterior margin and along the midline and single small denticles on the posterior margin. Anterior and posterior margins with few setae bent toward the genital pore. 

*Cinara (Schizolachnus) pineti* (Fabricius, 1781).

(Figure 6g–i).

**Abdominal tergite VIII:** possesses transverse rows of serrated imbrications and many long setae bent backward.

**Cauda:** short, triangular but slightly rounded, covered with slightly developed serrated imbrications, more prominent on the ventral side. Many long setae on the ventral side are bent toward the long axis of the body. 

**Anal plate:** wide, semi-circular, covered with short but sharp cuticular denticles near the anal pore and along the midline, and less prominent and sparsely arranged short rows of denticles or laterally serrated imbrications. Many long setae are directed toward the anal pore or bent toward the long axis of the body. 

**Genital plate:** wide, smooth with barely marked transverse rows of serration and several long setae on the posterior margin near the genital pore. 

*Tuberolachnus salignus* (Gmelin, 1790).

(Figure 6j–k).

**Abdominal tergite VIII:** covered with transverse rows of cuticular serration and many long setae directed backward. 

**Cauda:** very short, semilunar, dorsally covered with transverse rows of cuticular serration and ventrally with longer, sharp denticles closer to the anal pore and many long setae directed backward. 

**Anal plate:** wide, covered with short, serrated imbrications, more prominent closer to the anal pore, and many long setae. 

**Genital plate:** wide, smooth with barely marked transverse rows of serration and single denticles and many long setae on the posterior margin near the genital pore. 

*Eulachnus nigricola* (Pašek, 1953).

(Figure 6l).

**Abdominal tergite VIII:** covered with barely marked transverse rows of cuticular serration and several short setae ended with small clubs directed backward. 

**Cauda:** short, semicircular, dorsally covered with transverse rows of cuticular serration and ventrally with longer, sharp denticles closer to the anal pore and many long setae bent toward the long axis of the body. 

**Anal plate:** wide, covered with short, serrated imbrications, more prominent closer to the anal pore and many long setae bent toward the long axis of the body. 

**Genital plate:** rounded, smooth with barely marked transverse rows of serration and single denticles near the genital pore and many rather short setae bent toward the pore.

**Figure 6 insects-14-00471-f006:**
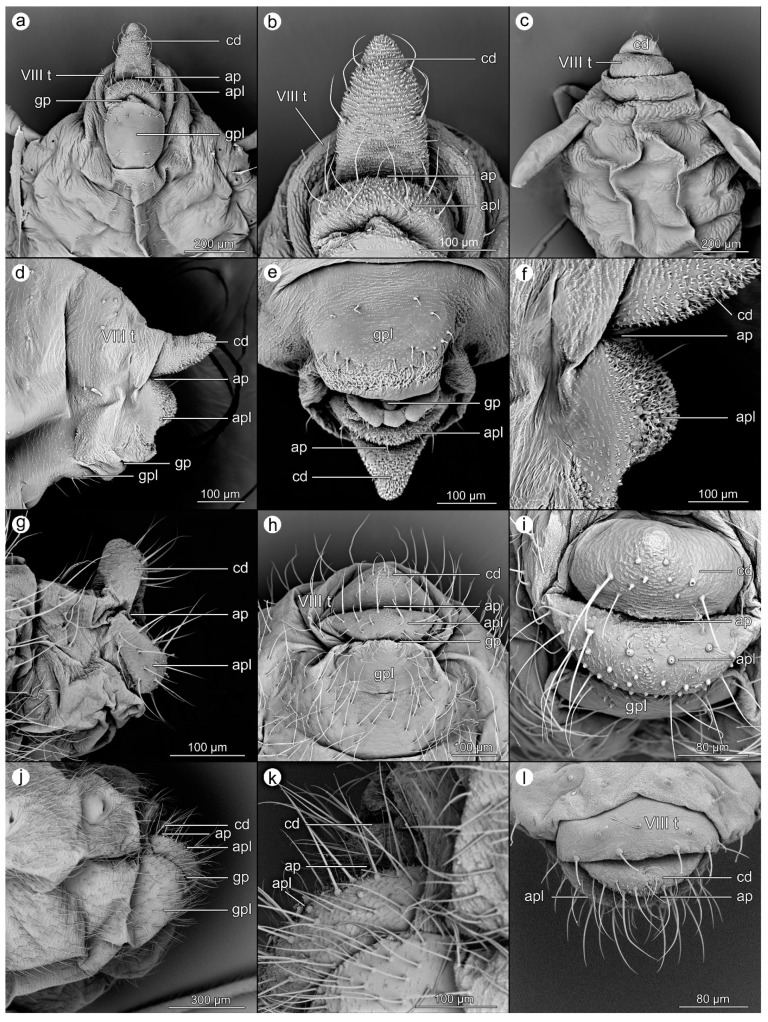
Perianal structures of *Liosomaphis berberidis*: (**a**)—ventral view, (**b**)—close-up on the surface of cauda and anal plate, (**c**)—dorsal view; *Brevicoryne brassicae*: (**d**)—lateral view, (**e**)—ventral view, (**f**)—close-up on the surface of cauda and anal plate; *Cinara (Schizolachnus) pineti*: (**g**)—lateral view, (**h**)—ventral view, (**i**)—rear view; *Tuberolachnus salignus*: (**j**)—lateral view, (**k**)—close-up on the surface of cauda; and *Eulachnus nigricola*: (**l**)—dorsal view.

### 3.2. Comparison of Non-Myrmecophilous and Myrmecophilous Aphids

The values of measured ratios in non-myrmecophilous aphids are presented in Table 1. The comparison of these values with myrmecophilous species (17 obligatory myrmecophilous and 12 facultatively myrmecophilous) with the application of a *t*-test under *p* < 0.05 indicated two ratios being significantly different. These differences are clearly presented in Figure 7, where these three groups of species are separated into two clusters with clearly different trend lines. The apl/apw ratio was significantly different between non-myrmecophilous and both facultatively and obligatory myrmecophilous species (*p* < 0.001), and the apl/cl ratio between non-myrmecophilous and also both facultatively and obligatory myrmecophilous species (*p* < 0.005). Additionally, the cw/chw ratio significantly differed between non-myrmecophilous and facultatively myrmecophilous aphids (*p* < 0.05).

The difference in results indicates the anal plate being more elongated in myrmecophilous aphids (apl/apw > 1.0) and the cauda being shorter than the length of the anal plate (apl/cl > 1).

**Figure 7 insects-14-00471-f007:**
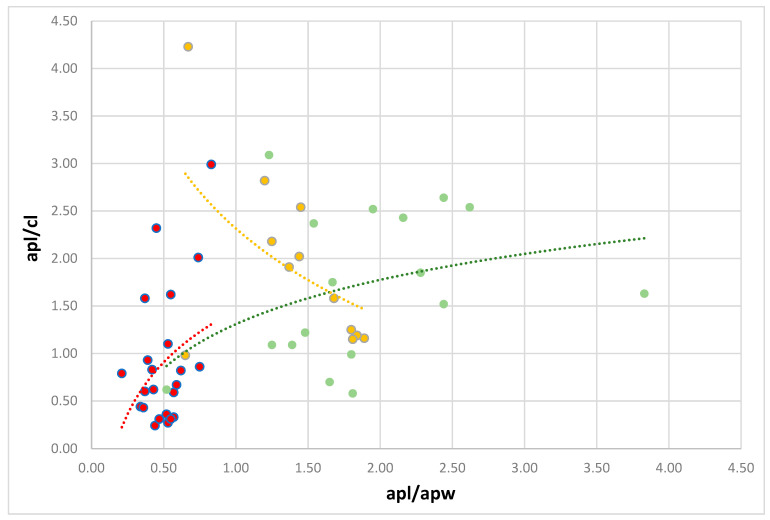
Differences in studied ratios between three groups of species: non-myrmecophilous (red), facultatively myrmecophilous (orange), and obligatory myrmecophilous (green). The dotted lines represent logarithmic trend lines. Abbreviations: apl/cl—anal plate length to cauda length ratio, apl/apw—anal plate length to anal plate width ratio. Data concerning facultatively and obligatory myrmecophilous species according to Supplementary Table S2 in [1].

## 4. Discussion

First of all, one should start by explaining the accepted standards for the species shown in the present paper as non-myrmecophilous. Species were selected for study as non-myrmecophilous based on both literature data and observations of all authors of the manuscript. Most of them were identical, although the authors included three species in this manuscript: *T. salignus, E. nigricola,* and *C.* (*S.*) *pineti*. Although some literature premises [41] may suggest their facultativeness, our observations show that ants, if present, are only occupying the same niche, without interspecific interaction in the manner characteristic of a mutualistic relationship. Therefore, it is very important to observe the colony because, looking at the density of ants, it is likely that they will always share the area with aphids, despite the lack of myrmecophily on their part (lack of signals such as care and protection of aphid colonies, reception of honeydew, lack of predatory behavior).


**Perianal structures: cauda and anal plate**


In this work, aphid species represent many types of cauda shapes previously described in publications and studies [42,43,44], such as finger-like with constriction (e.g., *Liosomaphis berberidis*, *Brevicoryne brassicae*), elongate (e.g., *Macrosiphoniella artemisiae*, *M. pulvera*, *Uroleucon aeneum*, *U. jaceae),* triangular (e.g., *Cinara* (*Schizolachnus*) *pineti*), semicircular (e.g., *Tuberolachnus salignus*, *Baizongia pistaciae*, *Eriosoma ulmi*, *Mimeuria ulmiphila*, *Tetraneura ulmi*), and knob-shaped (e.g., *Tinocallis takachihoensis. Myzocallis carpini, M. coryli*).

The genera *Macrosiphoniella* and *Uroleucon* (Figure 1 and Figure 2a–c) are characterized by very large, elongated caudas, which would seem to confirm the theory that the short and rounded shape of the cauda is typical of aphids visited by ants [22,45] so that an ant easily can reach the anus from which the aphid excretes the honeydew. However, this claim seems to be already doubtful with the species presented in Figure 2, with the observation of *Corylobium avellanae* and *Cavariella pastinacae*, whose caudas are relatively small, broadly finger-like, with an average length, which, in conjunction with the other species studied in this manuscript (e.g., *Clethrobius comes*), seems to quite confidently reject the length and shape of the cauda as a morphological feature that may be crucial in assessing the myrmecophily of a given aphid species. On the contrary, as shown before [1], many myrmecophilous aphids—e.g., of the genus *Aphis*—do have elongated cauda, which does not inhibit imbibing the droplet of honeydew by ants.

However, the length ratio of cauda to anal plate seems to be a significant marker of the involvement of aphids in mutualism with ants, with the special importance of the length of the anal plate. It seems that the structuring and morphology of the anal plate is a crucial adaptation for myrmecophily. As presented in Figure 7, non-myrmecophilous and myrmecophilous aphids form two clusters, relatively well separated and following the two important morphological features: the length of the anal plate and the length of the cauda in relation to the length of the anal plate. It seems that myrmecophilous aphids may have long cauda, but it must be shorter than the height of the anal plate. Furthermore, in the trophobiotic organ, the anal plate must be longer than its width. This finding partly conforms to the traditional view of the so-called “trophobiotic organ” with the reservation that it is not about the absolute length of the cauda (its shortening) but its relative shortness towards the anal plate and the length of the anal plate itself. Lachninae and Eriosomatinae have short or very short cauda and comprise non-myrmecophilous as well as myrmecophilous species. While this pattern is well supported by the results in studied Lachninae, in the case of Eriosomatinae, it is not so obvious because non-myrmecophilous *E. ulmi* and *P. fraxinifolii* have longer anal plates than myrmecophilous *B. pistaciae* and *T. ulmi*. However, Eriosomatinae aphids are host alternating species, with generations exclusively living in galls on the primary host without ants (e.g., *B. pistaciae* in Fordini [46]), thus having special adaptations for the safe removal of honeydew droplets, and in this study, only morphs living in galls were analyzed. In the case of myrmecophilous *Prociphilus* spp., being ant-attended both on primary as well as on secondary hosts [47], the ratios meet the above requirements for a trophobiotic organ [1]. 

A separate issue is the positioning of the anal plate, which in the case of myrmecophilous aphids is regarded to be vertically positioned to keep the honeydew droplet in the proper position to be removed by an ant worker. In the current study, it was difficult to explore the positioning of the anal plate. However, previous research on subterranean living aphids or aphids conducting cryptic life mode clearly indicated the horizontal positioning of cauda and dorsal exposition of the anal pore [29]. In this case, it should be connected to these aphids’ life mode, where the honeydew droplet’s ventral positioning is secure against spilling or contaminating the ant chamber where the aphids live, though again, not in the case of aphids of the genus *Stomaphis*, which generally also conduct cryptic life mode [48] yet feed in a vertical position [21]. In *Stomaphis*, there are certain supporting structures (e.g., perianal tubercles) and also the constant presence of ants which instantly remove any honeydew droplet.


**Perianal structures: Cuticle structure and setae**


The number, arrangement, and length of setae on the cauda and anal plate seem to be very variable features, strongly depending on the general morphological traits of the studied subfamily. It is difficult to conclude whether there is any correlation between these features with myrmecophily. All studied aphids, whether myrmecophilous or not, do have setae around the anal pore. Their presence, together with the development of cauda, seems to be the leading and evolutionary old adaptation to keep the excreted droplet of honeydew far from the body. It should be assumed that the disposal of feces is crucial for survival, and myrmecophilous aphids cannot wait too long for their removal by ants, especially taking into account the physiology of sap-feeding, where the aphid needs to utilize a vast quantity of nutritional fluid. The lateral positioning of setae on cauda in the case of aphids with longer cauda may be explained by the need to stabilize the position of the droplet until it is removed. Some attention may be given to the setae being bent towards the anus, but this is not only characteristic of myrmecophilous but also of non-myrmecophilous species. The setae and the wax covering are important, especially for aphids living in galls, which is a confined space posing certain difficulties in the removal of honeydew [49]. Figure 5 shows aphid species whose whole or part of the cycle takes place in galls with varying degrees of closure, from full galls in *Baizongia pistaciae* (large, horn-shaped galls on *Pistacia* trees) or *Tetraneura ulmi*, through wrinkled leaves galls in *Eriosoma ulmi* (yellowish or whitish green galls on elm), to *Mimeuria ulmiphila* living in terminal leaf nests formed on field maple (*Acer campestre*) by the inhibition of shoot growth, twisting and folding of leaves. Each of these species produces long wax strands from wax glands that protect their bodies. Although the amount of wax in individual species varies, it is always present in the perianal structure to protect the body from sticky honeydew (e.g., from clogging the spiracles). Of course, in closed galls, there is no possibility for ants to visit and collect honeydew, while in open galls, made of folded leaves, some species are facultatively myrmecophilic, such as *Dysaphis plantaginea*—where their body is covered with a delicate wax that does not form braids, but only a thin coating. As we can see in each of the galling species (Figure 5), there is a strongly shortened cauda, and the setae are shortened and not so densely arranged. Looking at galling aphids in lateral view (Figure 5a,d,g,j), we notice that the cauda never protrudes beyond the perianal structures; it is always aligned with the anal plates and sometimes even withdrawn into the body (*B. pistaciae*).

Finally, it seems that the cuticle and its microscuplture may play an auxiliary role in keeping honeydew droplets away from the aphid body. The mostly ventral surface of the cauda and part of the anal plate proximal to the anal pore are densely covered with cuticular denticles and spinules. Again, it is difficult to trace any correlation between myrmecophily and the robustness of the cuticular microscuplture. Some myrmecophilous species do have very prominent cuticular denticles and spinules, especially within the subfamily Aphidinae, but this cannot be generalized. For example, *Cinara pini* has very peculiar, mushroom-like spinules, but myrmecophilous *Lachnus pallipes* has a very weak microscuplture of the anal plate. Similarly, obligatory myrmecophilous *Stomaphis* has an almost smooth anal plate, but it also has a pair of laterally positioned tubercles densely covered by protuberant, sharp spinules [21]. In Eriosomatinae, we can also observe a very smooth surface of the anal plate regardless of the myrmecophily (Figure 5).

## 5. Conclusions

The performed studies showed that the concept of the trophobiotic organ in aphids as an adaptation to myrmecophily is true but requires revision. The new concept of the trophobiotic organ indicates an anal plate as long or longer than the cauda and as long or longer than its width. From morphological attitudes, it denotes anal plate length/cauda length ratio >1 and anal plate length/anal plate width (the widest part transverse to the long axis of the body) ratio >1. However, in the case of gall-living and host alternating Eriosomatinae, the issue of the role and evolution of perianal structures requires further studies.

## Figures and Tables

**Table 1 insects-14-00471-t001:** Ratio measurements in aphid species: apl/cl—anal plate length to cauda length ratio, apl/apw—anal plate length to anal plate width (the biggest width) ratio, cl/cw—cauda length to cauda width (at the base) ratio, cw/chw—cauda width (at the base) to cauda half width ratio.

	Subfamily	Species	apl/cl	apl/apw	cl/cw	cw/chw
1.	Drepanosiphinae	*Drepanosiphum platanoidis*	0.67	0.59	0.94	2.92
2.	Calaphidinae	*Clethrobius comes*	0.60	0.37	0.82	0.53
3.		*Myzocallis coryli*	0.33	0.57	0.86	0.85
4.		*Myzocallis carpini*	0.82	0.62	0.94	1.89
5.		*Tinocallis takachihoensis*	0.83	0.42	0.37	1.57
6.		*Euceraphis betulae*	0.59	0.57	1.22	2.46
7.	Eriosomatinae	*Mimeuria ulmiphila*	2.32	0.45	0.24	1.46
8.		*Tetraneura ulmi*	1.62	0.55	0.49	1.84
9.		*Eriosoma ulmi*	2.99	0.83	0.33	1.63
10.		*Baizongia pistaciae*	1.10	0.53	0.40	1.46
11.		*Prociphilus fraxinifolii*	2.01	0.74	0.43	1.66
12.	Lachninae	*Cinara* (*Schizolachnus*) *pineti*	0.93	0.39	0.57	1.47
13.		*Tuberolachnus salignus*	1.58	0.37	0.36	1.31
14.		*Eulachnus nigricola*	0.79	0.21	0.30	1.53
15.	Aphidinae	*Brevicoryne brassicae*	0.62	0.43	0.96	1.76
16.		*Cavariella theobaldi*	0.44	0.34	1.46	1.15
17.		*Cavariella pastinacae*	0.43	0.36	1.76	1.11
18.		*Corylobium avellanae*	0.86	0.75	1.04	0.74
19.		*Liosomaphis berberidis*	0.62	0.43	1.28	1.25
20.		*Uroleucon cichorii*	0.24	0.44	2.63	1.61
21.		*Uroleucon aeneum*	0.36	0.52	2.32	1.70
22.		*Uroleucon jaceae*	0.27	0.53	3.27	1.20
23.		*Macrosiphoniella artemisiae*	0.31	0.47	2.62	1.20
24.		*Macrosiphoniella pulvera*	0.36	0.52	2.93	1.08
25.		*Macrosiphoniella tanacetaria*	0.31	0.55	2.62	1.32
		mean	0.88	0.50	1.25	1.47

## Data Availability

All data generated or analyzed during this study are included in this published article.

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
