# Peer review of "Perianal Structures in Non-Myrmecophilous Aphids (Hemiptera, Aphididae)"

_insects, 2023, doi:10.3390/insects14050471_

Round 1

Reviewer 1 Report

There are some irregular style of the writing name of species, it should be corrected.

There some inconsistency/unclearanca about number of the population used, hust offered to check.

These are marked on manuscript

Author Response

All the authors of the article thank you very much for all valuable comments on the Manuscript, their approval certainly contributed to the improvement of the paper.

Reviewer 2 Report

Kaszyca–Taszakowska et al. gave detailed morphological descriptions of the perianal structures of 21 non-myrmecophilous aphid species from 5 subfamilies and provided beautiful SEM pictures. By statistical analysis of the perianal structure measurement data of 25 non-myrmecophilous, 17 obligately myrmecophilous, and 12 facultatively myrmecophilous aphid species, they found longer anal plates in myrmecophilous aphids, and the length and shape of cauda, cuticle structure, and setae seemed not to be the typical perianal structures correlated with myrmecophily. The definition of trophobiotic organ may need revision. This study provided valuable information about the perianal structure morphology of non-myrmecophilous aphids, and the results are of significance for a better understanding of morphological adaptations of aphids to mutualism with ants.

Major comments:

L153: The precondition of using t-test is that the data is normally distributed. Therefore, the distribution of data should be tested first. In addition, what software was used to produce Figure 7 should be provided in the Material and methods section.

L157-159: The perianal structures of 21 not 25 non-myrmecophilous aphid species were described in detail and corresponding SEM pictures were provided. Measurement data of 25 non-myrmecophilous species were used in the following statistical analysis. These should be stated clearly here.

L160: The species order in “3.1. Morphological characteristics of perianal structures” is confusing. They are not arranged according to subfamily or alphabetical order of species name. It would be better to list the descriptions of perianal structures according to the species order in Table 1.

L318, 349-350: “At the lower margin of the anal pore”. Is it “anal pore” here? Should it be “genital pore”? Besides, I suggest marking the obvious rudimentary gonapophyses in the figures.

L483: In the “3.2. Comparison of non-myrmecophilous and myrmecophilous aphids” result section, I would like to suggest displaying the visualization result of measurement data first, showing apl/apw>1.0 and apl/cl>1.0 in myrmecophilous aphids. Then talk about the statistical analysis results, showing there are significant differences in measurement data between non-myrmecophilous and myrmecophilous aphids.

L494-496: “Cauda width” and “anal plate width” should be defined clearly here. Is it width at the base or the biggest width?

L542It would be better to use “the length ratio of anal plate to cauda” here, which is consistent with the data used in Table 1.

L545-548: I suggest using apl/cl and apl/apw, which are presented in Figure 7, as “the two important morphological features” here. These two ratios actually represent the length and shape of anal plate. In other words, this study found that the key adaptive morphological character of myrmecophilous aphids was anal plate other than cauda which was considered important in previous hypothesis. Current statements of L545-548 are provided from the perspective of cauda, which is kind of not direct enough in logical. I suggest directly highlighting the importance of anal plate first, then talking about the cauda which seems to be not significantly correlated with myrmecophily.

L591-607: I did not get the main idea of this part about galling aphids. What I concerned about is how to tell whether the perianal structures of three galling species are correlated with non-myrmecophily or correlated with gall-living. In my opinion, these gall-living aphids cannot be considered as representatives of non-myrmecophilous species, as morphological adaptations to special gall environment may have taken place as well in these species.

L509: In the “Discussion” section, the authors focused on whether several perianal structures are significantly correlated with myrmecophily. I would like to suggest adding a conclusion paragraph at the end, where a modified definition of trophobiotic organ should be given. If it is difficult to give a definition now, it is better to clearly say that the definition needs to be revised. Whatever, the definition of trophobiotic organ is a key issue in this study, so the definition used currently should be stated in the Introduction and which morphological characters are regarded as trophobiotic organ should be listed too.

Minor comments:

L9-10: “we focused on the thoracic segment VIII, cauda and anal plate, their microstructure and setae.” “Genital plate” should be added here.

L20: “it is a crucial symbiosis enhancing their survival capability”. “Symbiosis” is not proper to describe the relationship between aphids and ants, “association” is more exact.

L70: “Internal symbionts” should be replaced with “endosymbionts”.

L74-98: “In view of the above-mentioned mutual influences in ant-aphid interactions …” It would be better to put this part in a separate paragraph.

L116-118: All of the 25 aphid species studied here are non-myrmecophilous. This should be stated clearly here in the “2.1. Taxon sampling” section, not only mentioning the Eriosomatinae species living in galls.

L122: Replace “Systematic order” with “Taxonomic system”.

L127: “Scanning electron microscopy was used as the primary method in this study.” It is better to add “(SEM)” after “Scanning electron microscopy”.

L131: What dose “PTA” mean? It only appears once throughout the ms. I suggest using the full name instead of an abbreviation.

L144-145: The abbreviations used here are not consistent with that used in the figures. In the figures, it is “VIII t” not “VIIIt”, “cd” not “cont.”. In addition, in “genital pore (gp) between them”, what does “them” refer to?

L151-152: “the source data on perianal structures presented by Kaszyca-Taszakowska et al. (2022) were applied (Fig. 7.)” I suggest modifying this sentence like “the source data on perianal structures of 17 obligatory myrmecophilous and 12 facultatively myrmecophilous aphid species presented by Kaszyca-Taszakowska et al. (2022) were applied (Fig. 7.)”.

L165: I suggest using “ventral view” instead of “abdominal view” in all figure legends.

L178: “Macrosiphoniella pulvera” in italic.

L303-305: Are “h” and “e” in the plate of Figure 4 labeled reversed?

L326, 339: Figure numbers of Prociphilus fraxinifolii and Tinocallis takachihoensis in the text seem wrong.

L356: Genital plate can be marked in Figure 5a.

L418: According to text, Figure 6l belongs to Eulachnus nigricola. Please modify the legend.

L501: It should be “apl/cl > 1”.

L507: It should be “Supplementary Table S1”. And it would be better to add a table title in the supplementary word file.

L509: I suggest adding subheadings in the Discussion section.

L537: I suggest providing the figure number of SEM picture of Clethrobius comes and describing its anal plate (i.e., short or long, shape) here.

L562-564: “which in the case of myrmecophilous aphids is regarded to be positioned vertically to keep the honeydew droplet in the proper position to be removed by ant worker.” Citation should be provided.

L603: “As we can see in each of the gall species”; L604: “Looking at gall aphids in lateral view”. Replace “gall” with “galling”.

Minor editing of English language required.

Author Response

(The authors gave the same response as above.)

Reviewer 3 Report

Dear Authors,

The research is undoubtedly well designed, articulated and supported with excellent images and research methodology. From this point of view. I do not have much to observe. The only problem I encountered is the English form which, in my opinion and advice, needs revision. Although the grammar is correct, many sentences are difficult to understand. The same words are often used twice in a sentence, there is little synthesis and poor use of commas. The latter would be useful to make the concepts expressed better understood. I recommend revision by a native English speaker. Beyond that, the research definitely deserves to be accepted and published.  

Best regards

The quality of English, in my opinion and advice, should be improved. Although the English form and grammar is correct, many sentences are difficult to understand, the same words are often used in the same sentence, there is little synthesis and poor use of commas. The latter would be useful to make the concepts expressed better understood. I recommend revision by a native English speaker. Beyond that, the research definitely deserves to be accepted and published.  

Author Response

The authors are very grateful for valuable comments on the Manuscript. We have revised the paper as suggested and believe it has added value and clarity as a result.

Round 2

Reviewer 2 Report

The authors have addressed most comments in the first round of review. But there are still several places need to be modified to improve the ms.

Major comments:

L173: 3.1. Morphological characteristics of perianal structures

I accept the authors’ response on why the species were listed in such an order. I suggest saying clearly about the reason of species list order here, which would be helpful for the readers to get the change trend of the “trophobiotic organ” as the authors said in the response.

L649: Conclusions.

The authors added a conclusion about the concept of the trophobiotic organ as I suggested. This is good. But I also suggested “the definition used currently should be stated in the Introduction and which morphological characters are regarded as trophobiotic organ should be listed too”. Please complement this content in the introduction.

Minor comments:

L160-162: Revise the sentence as follows, “Also, for comparison of non-myrmecophilous and myrmecophilous aphids, the source data on perianal structures of 17 obligatory myrmecophilous and 12 facultatively myrmecophilous aphid species presented by Kaszyca-Taszakowska et al. [1] were applied (Fig. 7).”

L164-165: “For statistical analysis of the obtained values the Shapiro-Wilk test for distribution type was performed”. The result of Shapiro-Wilk test should be provided here.

L167: “Statsoft Statistica 13.3 software”. Provide the citation of this software here.

L518-519: “cl/cw – cauda length to cauda width ratio”. Define what width of cauda.

L654: “anal plate length/ anal plate width ratio >1”. Define the anal plate width clearly, which is “the biggest width”.

Author Response

Thank you for your comments on the Manuscript, they have been included in the paper.
